# Immunomodulatory Effect of Hispolon on LPS-Induced RAW264.7 Cells and Mitogen/Alloantigen-Stimulated Spleen Lymphocytes of Mice

**DOI:** 10.3390/pharmaceutics14071423

**Published:** 2022-07-06

**Authors:** Eun Kyeong Lee, Eun Mi Koh, Yu Na Kim, Jeongah Song, Chi Hun Song, Kyung Jin Jung

**Affiliations:** 1Immunotoxicology Research Group, Korea Institute of Toxicology, 141 Gajeong-ro, Yuseong-gu, Daejeon 34114, Korea; eunkyeong.lee@kitox.re.kr (E.K.L.); emkoh@kitox.re.kr (E.M.K.); yuna.kim@kitox.re.kr (Y.N.K.); angryox@kitox.re.kr (C.H.S.); 2Animal Model Research Group, Korea Institute of Toxicology, Jeongeup 56212, Korea; jasong@kitox.re.kr

**Keywords:** ROS/RNS, hispolon, inflammation, immunomodulation, macrophage, mouse splenocyte, proliferation

## Abstract

Hispolon is a potent anticancer, anti-inflammatory, antioxidant, and antidiabetic agent isolated from *Phellinus linteus*, an oriental medicinal mushroom. However, the immunomodulatory mechanisms by which hispolon affects macrophages and lymphocytes remain poorly characterized. We investigated the immunomodulatory effects of hispolon on oxidative stress, inflammatory responses, and lymphocyte proliferation using lipopolysaccharide (LPS)-treated RAW264.7 macrophages or mitogen/alloantigen-treated mouse splenocytes. Hispolon inhibited LPS-induced reactive oxygen and nitrogen species (ROS/RNS) generation and decreased total sulfhydryl (SH) levels in a cell-free system and RAW264.7 cells. Hispolon exerted significant anti-inflammatory effects by inhibiting production of the proinflammatory cytokines interleukin 6 (IL-6) and tumor necrosis factor alpha (TNF-α) and activation of nuclear factor kappa B (NF-κB) and signal transducer and activator of transcription 3 (STAT3) in LPS-treated RAW264.7 cells. Hispolon also modulated NF-κB and STAT3 activation by suppressing the NF-κB p65 interaction with phospho-IκBα and the STAT3 interaction with JAK1, as determined via coimmunoprecipitation analysis. Additionally, hispolon significantly decreased lymphocyte proliferation, T cell responses and T helper type 1 (Th1)/type 2 (Th2) cytokines production in mitogen/alloantigen-treated splenocytes. We conclude that hispolon exerts immunomodulatory effects on LPS-treated macrophages or mitogen/alloantigen-treated splenocytes through antioxidant, anti-inflammatory, and antiproliferative activities. Thus, hispolon may be a therapeutic agent for treating immune-mediated inflammatory diseases.

## 1. Introduction

Hispolon (6-(3,4-dihydroxy-phenyl)-4-hydroxy-hexa-3,5-dien-2-one; C_12_H_12_O_4_) is a bioactive compound extracted from *Phellinus linteus* (Figure 1A), a mushroom [1] that has been reported to improve health and prevent various diseases, including diabetes, gastrointestinal diseases, allergies, and cancer [2,3,4,5]. Hispolon is known to possess biological activities, such as anticancer, antioxidant, antiviral, anti-inflammatory, immunomodulatory, and antidiabetic activities [6]. As previously reported, hispolon has the potential to induce apoptosis and cell cycle arrest and halt the metastasis of various cancer cells, including promyelocytic leukemia cells [3], glioblastoma cells [7], and lung cancer cells [8]. Hispolon targets the nuclear factor kappa B (NF-κB), mitogen-activated protein kinase (MAPK), and phosphatidylinositol 3-kinase (PI3K)/protein kinase B (Akt) signaling pathways to exert its anticancer effects [9,10]. However, the mechanism underlying the effect of hispolon on immunological responses remains poorly defined.

Inflammation is part of the body’s defense mechanism that eliminates injurious agents and promotes tissue repair/wound healing after injury or infection. The inflammatory process quickly ends under normal circumstances, but persistent and sometimes excessive stimulation of the inflammatory response is closely associated with various acute or chronic inflammatory diseases, such as infection, allergic reaction, arthritis, asthma, inflammatory bowel diseases, allograft rejection, and graft versus host disease (GVHD) [11,12,13,14,15]. These inflammatory responses are triggered by the immune system, which is divided into two types: the innate and adaptive immune systems. The innate immune system is the first line of host defense and nonspecifically eliminates invading pathogens through processes mediated by leukocytes such as monocytes/macrophages, granulocytes, mast cells, and dendritic cells. Once activated, these cells contribute to tissue inflammation by secreting proinflammatory cytokines (tumor necrosis factor (TNF)-α, interleukin (IL)-6, and IL-12), and chemokines and producing reactive oxygen and nitrogen species (ROS/RNS), leading to further activation of adaptive immunity. The adaptive immune system, comprising lymphocytes including T and B cells, exhibits a more efficient antigen-specific response to pathogens and establishes immunological memory [16]. The activation and proliferation of T and B cells are mandatory for adaptive immune system function, but the nonspecific, uncontrolled activation of these lymphocytes is correlated with inflammation-related autoimmune diseases such as inflammatory arthritis, systemic lupus erythematosus, and Crohn’s disease [17]. Additionally, many cancers arise from sites of infection, chronic irritation, and inflammation, suggesting that the tumor microenvironment is expansively integrated by inflammatory cells and signaling molecules of the immune system in the neoplastic process, promoting the proliferation, survival, and migration of tumor cells [18].

The inflammatory response is tightly regulated by several mediators and regulators released from inflammatory cells, including complement factors, chemokines, cytokines, eicosanoids, free radicals, and growth factors. It begins with the coordinated activation of signaling pathways that regulate inflammatory mediator levels [19].

The generation of ROS/RNS by phagocytic leukocytes, such as neutrophils, monocytes, macrophages, and eosinophils, plays a key role and serves as a hallmark of signaling pathways involved in the inflammatory process [20]. ROS/RNS participate in a variety of cellular stress mechanisms by damaging biological molecules such as proteins, lipids, and DNA [21]. In macrophages, nitric oxide (NO) formation is associated with the expression of the proinflammatory gene inducible nitric oxide synthase (iNOS) in response to cytokines, bacterial lipopolysaccharide (LPS), or parasites [22].

After oxidative challenge, the expression of proinflammatory genes is mediated by the activation of the redox-sensitive transcription factor NF-κB in most cells. NF-κB serves as a master regulator of immune and inflammatory responses [23]. NF-κB is activated by two distinct signaling pathways: the canonical and noncanonical pathways. Of these two signaling pathways, the canonical NF-κB pathway is involved in most immune responses. NF-κB consists of five different members, p65 (RelA), RelB, cRel, p50/p106 (NF-κB1), and p52/p100 (NF-κB2), and these proteins interact to form various homo- or heterodimers. The primary mechanisms of canonical NF-κB signaling are inhibitory kappa B kinase (IKK)-mediated phosphorylation, polyubiquitination, and subsequent degradation of inhibitor kappa B (IκB) proteins. NF-κB dimers are then released from cytoplasmic inhibition, translocate to the nucleus, bind DNA, and activate downstream gene transcription [24]. Another pathway that has been extensively reported to be involved in the dysregulated inflammatory response is the janus kinase/signal transduction and activator of transcription (JAK-STAT) signaling pathway. The JAK-STAT pathway regulates many cellular events, such as growth, survival, and differentiation, and is activated by a variety of ligands, including cytokines, hormones, and growth factors. The binding of ligands such as cytokines and growth factors to their corresponding receptors induces the activation of JAK, which induces tyrosine phosphorylation of STAT proteins, promoting their translocation to the nucleus and initiating the transcription of target genes [25]. Seven STAT proteins have been identified: STAT1, STAT2, STAT3, STAT4, STAT5 (STAT5α and STAT5β), and STAT6 [26]. Among the 7 STAT proteins, STAT3 activation plays important roles in cancer inflammation and immunity [27].

Based on the finding that LPS or concanavalin A (ConA) induces oxidative stress, inflammation, and proliferation of macrophages and splenocytes, much interest has been focused on how hispolon regulates the unnecessary activation of immune cells. The purpose of the present study was to define the antioxidant, anti-inflammatory, and antiproliferative activities of hispolon toward immune cells. Here, the immunomodulatory mechanisms of hispolon were explored in response to various conditions such as oxidative stress, pathogens, mitogens, and alloantigens.

## 2. Materials and Methods

### 2.1. Materials

2,2′-Azobis (2-methylpropionamidine) dihydrochloride (AAPH), 2-methyl-1,4-naphthoquinone (menadione), iron (II) sulfate (FeSO_4_), hydrogen peroxide (H_2_O_2_), LPS (Escherichia coli serotype O111:B4), 5,5′-dithiobis-(2-nitrobenzoic acid) (DTNB), N-acetyl-L-cysteine (NAC), esterase, ConA, mitomycin C, dexamethasone, and BCA protein reagent were obtained from Sigma–Aldrich (St. Louis, MO, USA). The Griess reagent kit and 2’,7’-dichlorodihydrofluorescein diacetate (DCFH-DA) were purchased from Molecular Probes (Eugene, OR, USA). Dulbecco’s Modified Eagle’s medium (DMEM), Roswell Park Memorial Institute (RPMI) 1640 medium, fetal bovine serum (FBS), and streptomycin-penicillin were purchased from GIBCO (Thermo Fisher Scientific, Inc., Waltham, MA, USA). The ONE-Glo^™^ luciferase assay system and FuGENE^®^ HD transfection reagent were purchased from Promega (Madison, WI, USA). Hispolon and primary antibodies against phospho-IκBα, IκBα, NF-κB p65, JAK1, phospho-JAK1, STAT3, and transcription factor (TF) IIB were obtained from Santa Cruz Biotechnology (Santa Cruz, CA, USA). Primary antibodies against phospho-STAT3 and α-tubulin and anti-rabbit secondary antibodies were purchased from Cell Signaling Technology (Danvers, MA, USA).

### 2.2. Animals

Six-week-old male BALB/c and DBA mice were obtained from Orient Bio Inc. (Seongnam-si, Korea) and maintained on a 12-h light–dark cycle in GLP-compliant facilities during the study. Animal testing was conducted in accordance with the guidelines of the American Association for Accreditation of Laboratory Animal Care (AAALAC, accredited since 1998) and the protocol approved by the Institutional Animal Care and Use Committee (IACUC) of the Korea Institute of Toxicology. 

### 2.3. Cell Culture

The RAW264.7 mouse macrophage cell line was acquired from the American Type Culture Collection (ATCC, Manassas, VA, USA), and the cells were maintained in DMEM supplemented with 10% heat-inactivated FBS, 100 U/mL penicillin, and 100 μg/mL streptomycin at 37 °C with 5% CO_2_. For splenocyte isolation, six male BALB/c and three male DBA mice (Orient Bio Inc., Seongnam-si, Korea) were used in this study. Fresh whole mouse spleen was carefully dissociated into a single-cell suspension using the rough end of a 10-mL syringe plunger. The cells were transferred to a 15-mL conical tube and centrifuged at 200× *g* for 10 min, and the supernatant was discarded. The cell pellet was vigorously resuspended in RPMI-1640 medium containing 10% heat-inactivated FBS, antibiotics (100 U/mL penicillin and 100 μg/mL streptomycin), and 2-mercaptoethanol, and incubated at 37 °C with 5% CO_2_.

### 2.4. Measurement of Cell Viability

For the cell viability assay, RAW264.7 cells were seeded into 96-well microplates and stabilized for 24 h. The cells were treated with various concentrations of hispolon for 6 or 24 h and then incubated for 2 h at 37 °C with Cell Counting Kit-8 (CCK-8) reagent (Dojindo Laboratories, Kumamoto, Japan). The absorbance was measured at 450 nm using a SpectraMax M3 microplate reader from Molecular Devices (Sunnyvale, CA, USA).

### 2.5. Measurement of ROS Levels

The in vitro radical-scavenging activity of hispolon was examined by measuring its superoxide and hydroxyl radical-scavenging activity. A mixture of FeSO_4_ (50 μM)/H_2_O_2_ (200 μM) and menadione were used as a source of hydroxyl radicals and superoxide, respectively. Various concentrations (1.25–10 μM) of hispolon were used in each experiment. DCFH-DA (20 μM) mixed with esterase (12 units/mL) was incubated with the sample in the dark at room temperature for 20 min. The fluorescence intensity of the samples was measured for 30 min using a SpectraMax M3 microplate fluorescence reader (Molecular Devices, Sunnyvale, CA, USA) with excitation and emission wavelengths of 485 and 535 nm, respectively.

The intracellular ROS levels were measured using the DCF-DA method to evaluate the antioxidative effect of hispolon in RAW264.7 cells. The cells were seeded in 96-well tissue culture plates at a density of 4 × 10^4^ cells/well and incubated for 24 h. Cells were pretreated with different concentrations of hispolon followed by stimulation with AAPH (400 μg/mL) and LPS (200 ng/mL). Finally, 25 μM DCFH-DA was added to each well, and the cells were incubated for 1 h.

The fluorescence intensity of the cells was detected using a SpectraMax M3 florescence microplate reader (Molecular Devices) with excitation and emission wavelengths of 485 and 535 nm, respectively. The intracellular ROS level was also cross-confirmed using the DCF-DA method with a FACSCalibur flow cytometer (BD Biosciences, San Jose, CA, USA).

### 2.6. Measurement of Total Sulfhydryl (SH) Levels

A mixture containing 250 μL of 0.2 M Tris buffer (pH 8.0), 25 μL of 0.01 M DTNB, 500 μL of methanol was added to 25 μL of cell lysates and incubated for 15 min at 23 °C to determine the total SH levels. After centrifugation at 600× *g* for 20 min, the absorbance of the supernatant was detected at 412 nm using a SpectraMax M3 microplate reader (Molecular Devices, Sunnyvale, CA, USA).

### 2.7. Griess Assay

The culture supernatants were mixed with equal volumes of Griess reagent and then incubated for 30 min to measure the amount of NO released by RAW264.7 cells. Absorbance was read at 540 nm using a SpectraMax M3 microplate reader (Molecular Devices, Sunnyvale, CA, USA).

### 2.8. Transient Transfection and Luciferase Reporter Assay

RAW264.7 cells were transiently transfected with FuGENE HD transfection reagent and plasmids encoding an NF-κB luciferase reporter (0.1 μg per well; Clontech Laboratories, Inc., Mountain View, CA, USA). Briefly, 4 × 10^4^ cells were plated in each well of 24-well plates. After an overnight incubation, the cells were transfected with 0.1 μg of DNA/0.3 μL of FuGENE HD reagent for 24 h. Subsequently, the transfected cells were stimulated with LPS for 6 h in the presence or absence of hispolon, washed with PBS, and then ONE-Glo luciferase assay reagent was added to the plate. Luciferase activity was measured using a luminometer (SpectraMax M3, Molecular Devices, Sunnyvale, CA, USA).

### 2.9. Preparation of Cell Lysates and Western Blot Analysis

Cells were rinsed with ice-cold PBS, followed by centrifugation. The collected cell pellets were lysed with NE-PER nuclear and cytoplasmic extraction reagents according to the manufacturer’s instructions (Pierce; Thermo Fisher Scientific, Inc., Waltham, MA, USA). After determining the concentration using the BCA protein assay, equal amounts of protein were loaded on an 8–12% (*w*/*v*) SDS-polyacrylamide gel, electrophoresed, and transferred to polyvinylidene fluoride. Each immunoblot was blocked and probed with primary antibodies, and secondary antibodies conjugated to horseradish peroxidase. Antibody labeling was visualized by detecting chemiluminescence using a SuperSignal West Pico PLUS ECL kit (Pierce; Thermo Fisher Scientific, Inc., Waltham, MA, USA).

### 2.10. Coimmunoprecipitation

The lysate was precleared to minimize nonspecific binding of proteins to G and L agarose beads, and protein complexes were formed in the lysate when samples were incubated overnight at 4 °C with a monoclonal anti-JAK1 or anti-NF-κB p65 antibody. The immune complexes were washed five times with lysis buffer. The immunoprecipitated proteins were eluted by boiling in SDS-containing buffer for 10 min. The eluted sample was assessed using SDS-PAGE. The interactions of JAK1 with STAT3 and NF-κB p65 with phospho-IκBα were measured by performing Western blot analysis.

### 2.11. Mitogenicity

Splenocytes from BALB/c mice were seeded in 96-well tissue culture plates at a density of 2.5 × 10^6^ cells/well. Splenocytes were pretreated with hispolon, and then ConA (T-cell mitogen, 1 μg/mL) or LPS (B-cell mitogen, 0.2 μg/mL) was added. After 72 h of incubation, the number of proliferating cells was measured using CCK-8 reagent. The optical density was measured at 450 nm using a SpectraMax M3 microplate reader (Molecular Devices, Sunnyvale, CA, USA).

### 2.12. Mixed Lymphocyte Reaction (MLR)

The MLR assay was performed using the method described by Itoh et al. (1993) [28], with modifications. Splenocytes from DBA mice at a density of 2 × 10^7^ cells/mL were treated with mitomycin C (200 μg/mL, 37 °C for 45 min). The cells were rinsed twice with RPMI 1640 medium containing 10% FBS and adjusted to a density of 4 × 10^6^ cells/mL (stimulator cells). A second set of splenocytes (responder cells) was prepared from BALB/c mice and adjusted to a density of 2 × 10^6^ cells/mL. Responder cells (BALB/c) were cocultured at a ratio of 1:2 with stimulator cells (DBA) in RPMI 1640 medium supplemented with or without hispolon in 96-well U-bottom culture plates. After 5 days of culture with 5% CO_2_ at 37 °C, cell proliferation was examined by adding CCK-8 reagent. The optical density was measured at a wavelength of 450 nm using a SpectraMax M3 microplate reader (Molecular Devices, Sunnyvale, CA, USA).

### 2.13. Cytokine Measurements

Commercially available IL-6 and TNF-α ELISA kits (BD Biosciences) were used to determine the IL-6 and TNF-α concentrations in the supernatant of RAW264.7 cells according to the manufacturer’s protocols.

Splenocytes from BALB/c mice were plated in 96-well plates at a density of 2.5 × 10^6^ cells/well. Supernatants from the cultured splenocytes were measured using a cytometric bead array (CBA) immunoassay with a mouse T helper type 1 (Th1)/type 2 (Th2)/type 17 (Th17) cytokine kit according to the manufacturer’s guidelines (BD Biosciences). The cytokine levels in the culture supernatants were detected using CBA mouse flex set capture beads, which consisted of a population of single beads with a distinct fluorescence intensity and a coating of specific antibodies recognizing detectable cytokines. The captured cytokines were detected with specific antibodies conjugated to PE dye. The samples were analyzed using a FACSCalibur flow cytometer (BD Biosciences) with FCAP array multiplex analysis software to calculate the cytokine concentrations and plot the standard curves.

### 2.14. Statistical Analysis

The results were analyzed statistically using one-way analysis of variance (ANOVA), and *p* < 0.05 was considered statistically significant.

## 3. Results

### 3.1. Effect of Hispolon on the Viability of RAW264.7 Cells

Before evaluating the biological effect of hispolon, cytotoxicity was first determined by treating cells with 0.62, 1.25, 2.5, 5, and 10 µM hispolon for 6 h or 24 h. Treatment with 0.62, 1.25, 2.5, 5, and 10 µM hispolon caused no cytotoxicity at 6 h (Figure 1B) or 24 h (Figure 1C). Thus, a concentration of hispolon less than 10 µM was used in all subsequent experiments.

### 3.2. In vitro and Cellular Antioxidant Activity of Hispolon

The in vitro ROS-scavenging ability of hispolon was investigated using the DCF-DA method to measure the levels of superoxide and hydroxyl radicals generated from menadione and hydrogen peroxide, respectively. As shown in Figure 2, hispolon dose-dependently decreased the levels of superoxide produced from menadione (Figure 2A) and hydroxyl radicals generated from hydrogen peroxide (Figure 2B). Based on these data, the scavenging activity of hispolon was as effective as that of the antioxidant NAC. AAPH was used as a positive control.

Next, RAW264.7 cells were preincubated with 0.62, 1.25, 2.5, 5, or 10 µM hispolon for 1 h and then treated with 400 µg/mL AAPH or 200 ng/mL LPS to evaluate the cellular antioxidant ability of hispolon. First, the intracellular ROS-scavenging effect of hispolon was determined in RAW264.7 cells with oxidative stress induced by AAPH or LPS. Hispolon decreased ROS generation induced by AAPH (Figure 2C) or LPS (Figure 2D) stimulation in a dose-dependent manner. Additionally, the intracellular ROS-scavenging effects of hispolon were confirmed using a flow cytometer, and the results in Figure 2E,F were consistent with those shown in Figure 2D. Second, we investigated the effect of hispolon on LPS-induced NO production in RAW264.7 cells. NO is a gaseous free radical with potent biological effects, but in most organisms, overproduction of NO is associated with a strong inflammatory process and damaged tissue, leading to the generation of highly RNS, which react with superoxide [22]. As shown in Figure 2G, hispolon significantly inhibited NO generation induced by LPS in RAW264.7 cells. Finally, total SH levels in LPS-treated RAW264.7 cells were detected to determine the effect of hispolon on thiol levels under conditions of oxidative stress. Thiols are organic compounds containing an SH group with a major role in preventing cellular oxidative stress as endogenous redox regulators [29]. The total SH levels induced by AAPH were significantly decreased compared with those in the nontreated control group, while hispolon increased the total SH levels in a dose-dependent manner (Figure 2H). The results shown in Figure 2 indicate that hispolon maintained the intracellular redox status by inhibiting ROS generation and increasing total SH levels in RAW264.7 cells.

### 3.3. Suppression of TNF-α and IL-6 Production by Hispolon in LPS-Induced RAW264.7 Cells

RAW264.7 cells were stimulated with LPS, and the production of the inflammatory cytokines, TNF-α and IL-6, was measured using ELISA kits to examine the effect of hispolon on the production of proinflammatory cytokines. As shown in Figure 3A,B, hispolon significantly suppressed LPS-induced production of TNF-α and IL-6 in a dose-dependent manner. Dexamethasone was used as a positive control.

### 3.4. Inhibition of NF-κB Activation by Hispolon in LPS-Stimulated RAW264.7 Cells

We assessed NF-κB activation using Western blotting analysis and reporter gene assays to investigate the molecular mechanism underlying hispolon-mediated inhibition of the inflammatory response. We first examined the nuclear levels of the NF-κB p65 protein and the cytosolic levels of the IκBα and phospho-IκBα proteins in LPS-stimulated RAW264.7 cells treated with or without hispolon. As shown in Figure 4A,B, the nuclear translocation of NF-κB p65 and the phosphorylation of cytoplasmic IκBα were increased by LPS, whereas hispolon significantly decreased their levels. The cytoplasmic IκBα levels were not changed by LPS or hispolon treatment. Second, luciferase activity was detected in cells transfected with NF-κB luciferase reporter vector to verify the effect of hispolon on the transcriptional activity of NF-κB in LPS-induced RAW264.7 cells. NF-κB luciferase activity was substantially increased following LPS treatment, and hispolon significantly suppressed this LPS-induced NF-κB activation (Figure 4C). Moreover, in Figure 4D, the coimmunoprecipitation analysis showed that hispolon inhibited the interaction between NF-κB p65 and phosphorylated IκBα. These findings suggest that hispolon exerted an anti-inflammatory effect by blocking NF-κB activation and markedly hindering NF-κB p65 translocation by inhibiting IκBα phosphorylation.

### 3.5. Inhibition of LPS-Induced JAK1/STAT3 Signaling by Hispolon

Abnormal activation of the JAK/STAT signaling pathway may result in cancer and immune system-related diseases [30]. IL-6 plays a major role in the activation of the JAK/STAT pathway [31]. As shown in Figure 3, the production of the proinflammatory cytokine IL-6 was attenuated by hispolon. Therefore, we investigated the pathway downstream of IL-6 by measuring the phosphorylation of cytoplasmic JAK1 and nuclear STAT3 using Western blotting analysis. The levels of phosphorylated JAK1 and STAT3 were increased by LPS stimulation, whereas hispolon reduced the increases in the phosphorylation of JAK1 in the cytoplasm and STAT3 in the nucleus in a dose-dependent manner. Total JAK1 and STAT3 levels were not changed by LPS or hispolon treatment (Figure 5A,B). Moreover, as shown in Figure 5C, the coimmunoprecipitation assay revealed that hispolon inhibited the interaction between JAK1 and STAT3. Based on these results, hispolon exerted to anti-inflammatory effects by inhibiting not only the phosphorylation of cytoplasmic JAK1 and nuclear STAT3 but also the JAK1-STAT3 interaction in LPS-stimulated RAW264.7 cells.

### 3.6. Modulation of Mitogen or Alloantigen-Stimulated Lymphocyte Proliferation by Hispolon

Common mitogens, i.e., ConA and LPS, were added to mouse splenocytes treated with or without hispolon to determine the effect of hispolon on mitogen-stimulated lymphocyte proliferation. In ConA- and LPS-treated groups, the proliferation of both T and B lymphocytes was substantially increased, while hispolon decreased the proliferation of both lymphocyte types in a dose-dependent manner (Figure 6A). MLR assays were performed to observe cell-mediated immune function. In these experiments, the splenocytes from two individual groups (responders and stimulators) were mixed in vitro. As shown in Figure 6B, the group treated without hispolon showed a considerable increase in lymphocyte proliferation, while hispolon decreased lymphoproliferative activity in a dose-dependent manner. The suppressive potency of hispolon was compared to that of the immunosuppressive agent dexamethasone, which served as a positive control. The data indicated that hispolon functioned as a potential suppressor of lymphocyte proliferation.

### 3.7. Suppression of Cytokine Production from ConA-Treated Splenocytes by Hispolon

A CBA assay of Th1/Th2/Th17 cytokines was conducted to measure the level of Th1 cytokines (IL-2, TNF-α, and IFN-γ), Th2 cytokines (IL-4, IL-6, and IL-0), and Th17 cytokines (IL-17A) and elucidate the effect of hispolon on cytokine production in ConA-treated mouse splenocytes. The production of IL-2, IFN-γ, TNF-α, and IL-6 was significantly higher in ConA-treated splenocytes than in the control group, while hispolon treatment decreased their production in a dose-dependent manner (Figure 7A–D). The levels of the remaining cytokines were not significantly altered by ConA or hispolon treatment (data not shown for IL-4, IL-10, and IL-17A).

## 4. Discussion

In the present study, we focused on examining the immunomodulatory effects and mechanism of action of hispolon on macrophage RAW264.7 cells and mouse splenocytes. We showed that hispolon exhibited antioxidant effects by decreasing ROS/RNS levels and increasing the total SH levels in LPS/AAPH-treated RAW264.7 cells. Additionally, hispolon exerted anti-inflammatory effects by significantly reducing the levels of the proinflammatory cytokines IL-6 and TNF-α and inhibiting the NF-κB p65/IκBα and JAK1/STAT3 signaling pathways in LPS-treated RAW264.7 cells. Furthermore, hispolon suppressed lymphocyte proliferation, T cell response, and Th1/Th2 cytokine production in mitogen- or alloantigen-stimulated mouse splenocytes. Therefore, our data are the first to show that hispolon exerts immunomodulatory effects on LPS-treated macrophages and mitogen/alloantigen-treated mouse splenocytes through its antioxidant, anti-inflammatory, and antiproliferative activities.

The present study focused first on assessing the antioxidant effects of hispolon. In this study, hispolon alleviated ROS/RNS generation in both cell-free systems and cell culture systems (Figure 2). In addition, hispolon increased the decreased total SH levels in AAPH-treated RAW264.7 cells. These observations are consistent with previous reports [32,33] that hispolon has the potential to function as a redox regulator, although the experimental model is different. Huang et al. [32] reported that hispolon protects against liver damage by reducing the levels of the oxidative stress markers malondialdehyde and NO, as well as increasing the activities of the antioxidant enzymes superoxide dismutase (SOD), catalase, and glutathione peroxidase in carbon tetrachloride (CCL4) treated rats. Huang et al. [33] also reported that the redox-regulating activity of hispolon is involved in molecular mechanisms protecting against lung injury in LPS-challenged mice. Redox regulation of the immune response is critically involved in physiological and pathological reactions [34,35]. A pro-oxidant environment in the initial stages of an immune response facilitates the phagocytic functions of innate immune cells, namely, macrophages and neutrophils, as well as the priming of T cells. However, excessive ROS generation and/or decreased activity of cellular defense systems leads to toxicity toward self-components; tissue or organ failure may subsequently occur, ultimately leading to immune dysregulation and inflammatory diseases, such as allergies, allograft rejection, autoimmune diseases, systemic lupus erythematous, and inflammatory bowel diseases [15,36,37,38]. Therefore, we suggest that hispolon might improve the dysregulation of inflammatory immune cells by modulating the cellular redox status.

Our data also showed that hispolon inhibits the LPS-mediated secretion of the proinflammatory cytokines IL-6 and TNF-α from RAW264.7 cells. IL-6 and TNF-α play critical roles in regulating of inflammatory responses and have been implicated in the pathogenesis of acute or chronic inflammatory diseases [39,40]. TNF-α is a potent inflammatory mediator that induces the chemotaxis of neutrophils and T lymphocytes. Additionally, TNF-α acts synergistically in NO production [41], and blockade of TNF-α has been shown to indirectly decrease NO production and iNOS expression [42] and improve impaired cellular defenses against pathogenic invaders [43]. IL-6 is a pleiotropic inflammatory cytokine that is produced by a variety of cells, including lymphocytes, macrophages, fibroblasts, and smooth muscle cells, and is involved in inflammation, hematopoiesis, and immune responses [44]. Importantly, TNF-α and IL-6 levels are tightly regulated by the transcriptional regulatory proteins NF-κB and STAT3. NF-κB-induced IL-6 secretion triggers a positive feedback loop of NF-κB and STAT3 activation [45]. The results shown in Figure 3, Figure 4 and Figure 5 indicate that hispolon suppresses STAT3 activation through the inhibition of NF-κB-induced IL-6 expression and prevents STAT3 activation by directly inhibiting the interaction of STAT3 with JAK1. In particular, we are the first to elucidate the molecular mechanism by which hispolon directly inhibits the interaction of STAT3 with JAK1 and the interaction of NF-κB p65 with p-IκBα.

NF-κB and STAT3 are important transcription factors linking inflammation to cancer [46]. Based on accumulating evidence, constitutive activation of NF-κB and STAT3 occurs in various cancer types, such as colon, gastric, lung, and liver cancers. NF-κB activation increases the expression of major inflammatory genes that promote the proliferation, invasion, angiogenesis, and metastasis involved in tumorigenesis [47]. STAT3 signaling also induces a cancer-promoting inflammatory environment by inducing the expression of inflammatory cytokines, chemokines, and other mediators [30]. Interaction between NF-κB and STAT3 in tumor microenvironment is known to increase tumors promoting inflammation and pro-tumor signaling to be polarized the M2 macrophage phenotype. In contrast, blockade of NF-κB and STAT3 is reported to reduce tumor escape from immunosurveillance, decreasing growth, and the spread of breast cancer [48]. Therefore, drugs targeting STAT3 and NF-κB, which lead to chronic inflammation, are considered an important strategy for both cancer prevention and therapy. According to our results, hispolon has the potential to suppress inflammatory responses by inhibiting the activation of both STAT3 and NF-κB and inducing the production of the proinflammatory cytokines, TNF-α and IL-6 in LPS-treated macrophages. These anti-inflammatory effects targeting transcription factors STAT3 and NF-κB and cytokine IL-6 on macrophage might contribute to the anti-tumor immune microenvironment by blocking the polarization of macrophages into M2. Recently, Lee et al., [49] reported that *Phellinus linteus* extract, which contains hispolon, increases the expression of M1 macrophages-related genes, including TNFα and IL-6 on THP-1 cells exposed with M1 stimulator. We have not confirmed whether hispolin affects M1 and M2 macrophage polarization, but further studies are needed to verify the mechanism of action of hispolon.

Hispolon, a natural polyphenol, is known to have potential cytotoxic activity against 13 different types of cancers and shows particularly excellent efficacy against breast cancer [6]. Hispolon induces cell cycle arrest and apoptosis and reduces metastasis by targeting multiple cellular signaling pathways, including the MAPK, PI3K/Akt, and NF-κB pathways [6], as well as mitochondrial and STAT3 pathways [50] in cancer cells. These reported data suggest the potential of hispolon as a potent multi-target anticancer drug. Chethna et al. [51] reported that hispolon showed higher toxicity in three types of cancer cells (MCF7, A549, and CHO) than splenic lymphocytes at the same dose. From the results depicted in Figure 1, the doses of hispolon used in our experiment (≤10 μM) exhibited anti-inflammatory effects but not a significant cytotoxic effect on macrophages. These cytotoxicity results indicate hispolon is selective towards tumor cells.

We further investigated the effect of hispolon on both mitogen and alloantigen-induced cell proliferation and Th1/Th2 cytokine production. As shown in Figure 6A, hispolon significantly suppressed responsiveness to the T lymphocyte mitogen ConA and the B lymphocyte mitogen LPS. Next, we examined the modulation of alloantigen-related proliferation by hispolon in splenocytes. T cell responses were measured using an MLR assay, in which stimulator cells recognize alloantigen on responder cells, to confirm the effect on alloantigen-related proliferation. As shown in Figure 6B, the proliferation of alloantigen-specific T cells was reduced by hispolon, and furthermore, hispolon suppressed the secretion of IL-2, IFN-γ, TNF-α (Th1-type cytokines), and IL-6 (Th2-type cytokine) by splenocytes, implying that hispolon efficiently modulated both Th1- and Th2-mediated immune responses (Figure 7A–D). Among the Th1/Th2 cytokines, IL-2 is an essential cytokine related to T-cell mediated acute transplant rejection [52]. Several studies have reported that treatment with anti-IL-2 receptor antibody is effective at reducing the rate of acute rejection in organ transplant recipients [53,54]. Therefore, the inhibitory effect of hispolon on lymphocyte proliferation, T cell responses, and Th1/Th2 cytokine production suggests the potential of hispolon to serve as a novel immunosuppressant.

In conclusion, our data show that hispolon exerts potent antioxidant effects on macrophages by decreasing in ROS/RNS production, increasing total cellular SH levels, and exerting anti-inflammatory effects by decreasing TNF-α and IL-6 production, and suppressing the activity of the NF-κB/IκBα and JAK1/STAT3 signaling pathways. Moreover, in splenic lymphocytes, hispolon exerts anti-proliferative and anti-inflammatory effects by inhibiting lymphocyte proliferation, T cell responses, and Th1/Th2-related pro-inflammatory cytokine production (Figure 1). Together, these findings reveal that hispolon exerts a powerful immunomodulatory effect and might be developed as an effective therapeutic agent for immune-mediated diseases characterized by inflammation.

## Data Availability

The data presented in this study are contained within the article.

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
