# Peer review of "Immunomodulatory Effect of Hispolon on LPS-Induced RAW264.7 Cells and Mitogen/Alloantigen-Stimulated Spleen Lymphocytes of Mice"

_pharmaceutics, 2022, doi:10.3390/pharmaceutics14071423_

Round 1

Reviewer 1 Report

This manucript has discussed the immunomodulatory effect of hispolon by inhibition of cytokine production, and suppression of the NF-κB and STAT3 signaling pathways. It has showed very impressive results and discussions.

It would be great to add the chemical strcuture of hispolon to a figure.

An overview scheme of the immunomodulatory effect of hispolon (or you can add information to figure 7E then move to discussion and conclusion part) to discussion and conclusion part would be helpful for readers to catch up. 

Did the author investigate the MAPK siganling pathway as it is also related to cancer? 

Add some references to L32 and L34. 

Some format issues have to imrpove.

Add full name before using the abbreviation, such as MAPK.

Add space before the reference, such as it should be antidiabetic [1] , not antidiabetic[1]. Add space between p<0.01 as well. 

L125, carried out in accordance

L151, delete one .

Add n = ? to all the figure legend. The author have to clarify how many mice used for experiment and this information have to add to Materials and Methods. 

Add Dexamethasone (Dexa) was used for positive control to figure 3 legend.

L462, Add more recent references with 38 and 39 

Reviewer 2 Report

In the manuscript, "Immunomodulatory effect of hispolon on LPS-induced 2 RAW264.7 cells and mitogen/alloantigen-stimulated spleen 3 lymphocytes of mice", Lee et al. investigated the effects of hispolon on myeloid mouse cell lines and splenocytes.

Pathways affected by hispolone and mechanisms, e.g. attenuation of LPS stimulation, that are presented in this study were already published in 2020 by others (see Reference 24: Huang et al. (2020). Attenuation of Lipopolysaccharide Induced Acute Lung Injury by Hispolon in Mice, Through Regulating the TLR4/PI3K/Akt/mTOR and 566 Keap1/Nrf2/HO-1 Pathways, and Suppressing Oxidative Stress-Mediated ER Stress-Induced 567 Apoptosis and Autophagy).

Thus, novelty and originality of their data are clearly missing. In addition, it is not evident why the authors have chosen to use mouse immune cell lines and splenocytes, instead of human immune cells (primary or cell lines) to show proof on concept of immunomodulatory effects of hispolon.

Taken together, the results provided in this work are no advancement of the current knowledge of hispolon effects.

Round 2

Reviewer 2 Report

Dear authors / dear editor

I have read your comments to my concerns regarding the current manuscript. Still I believe that “proof of principle” in the human system ex-vivo would clearly raise the impact of the current study.

However, I leave it to the editor to decide if the manuscript fulfills, in terms of novelty and interests to the readers, in its current form the scope of the journal.